# The Role of Cellular Senescence in Cyclophosphamide-Induced Primary Ovarian Insufficiency

**DOI:** 10.3390/ijms242417193

**Published:** 2023-12-06

**Authors:** Zixin Xu, Nozomi Takahashi, Miyuki Harada, Chisato Kunitomi, Akari Kusamoto, Hiroshi Koike, Tsurugi Tanaka, Nanoka Sakaguchi, Yoko Urata, Osamu Wada-Hiraike, Yasushi Hirota, Yutaka Osuga

**Affiliations:** Department of Obstetrics and Gynecology, Faculty of Medicine, The University of Tokyo, Tokyo 113-8655, Japan; xuzixin0128@gmail.com (Z.X.); yutakaos-tky@umin.ac.jp (Y.O.)

**Keywords:** cellular senescence, cyclophosphamide, primary ovarian insufficiency, senolytics, senotherapeutics

## Abstract

Young female cancer patients can develop chemotherapy-induced primary ovarian insufficiency (POI). Cyclophosphamide (Cy) is one of the most widely used chemotherapies and has the highest risk of damaging the ovaries. Recent studies elucidated the pivotal roles of cellular senescence, which is characterized by permanent cell growth arrest, in the pathologies of various diseases. Moreover, several promising senolytics, including dasatinib and quercetin (DQ), which remove senescent cells, are being developed. In the present study, we investigated whether cellular senescence is involved in Cy-induced POI and whether DQ treatment rescues Cy-induced ovarian damage. Expression of the cellular senescence markers p16, p21, p53, and γH2AX was upregulated in granulosa cells of POI mice and in human granulosa cells treated with Cy, which was abrogated by DQ treatment. The administration of Cy decreased the numbers of primordial and primary follicles, with a concomitant increase in the ratio of growing to dormant follicles, which was partially rescued by DQ. Moreover, DQ treatment significantly improved the response to ovulation induction and fertility in POI mice by extending reproductive life. Thus, cellular senescence plays critical roles in Cy-induced POI, and targeting senescent cells with senolytics, such as DQ, might be a promising strategy to protect against Cy-induced ovarian damage.

## 1. Introduction

Due to major treatment advances in recent decades, 85% of children with cancer now survive 5 years or more [1], and the 5-year relative survival rate of patients with breast cancer, the most commonly diagnosed cancer among American women, at an early stage is higher than 90% [2]. Therefore, quality of life after cancer treatment, a major component of which is fertility, has been emphasized.

Chemotherapy is commonly used to treat cancer. Chemotherapy-induced gonadotoxicity depends on the chemotherapy protocol, including the total administered dose and the age and ovarian reserve of the patient at the time of treatment [3]. Cyclophosphamide (Cy) is one of the most widely used chemotherapies for a wide range of cancers, including lymphomas, leukemia, neuroblastoma, retinoblastoma, and breast cancer, as well as autoimmune diseases. Alkylating agents including Cy have the highest risk of damaging ovaries. It was reported that 42% of women treated with alkylating agents developed ovarian insufficiency [4]. Primary ovarian insufficiency (POI), also known as premature ovarian failure and premature menopause, is a medical condition in which ovarian follicles are depleted and cease to function normally, as both reproductive and endocrine organs in women younger than 40 years [5]. The mechanisms underlying the development of POI not only involve the direct cytotoxic effect of chemotherapeutic agents; the disappearance of primordial follicles can also be due to the induction of dormant follicles into the growing stage, a phenomenon called ‘burn-out’ [6]. Several strategies to preserve fertility, such as ovarian tissue and embryo cryopreservation, have been proposed. However, there is no method to overcome the ovarian damage induced by chemotherapy. Therefore, new drugs must be explored.

Cellular senescence is a cellular state of stable and long-term loss of proliferative capacity but with the retention of normal metabolic activity and viability [6]. It is characterized by irreversible replicative arrest, activation of tumor suppressors, profound chromatin changes, and resistance to apoptosis. In addition, cellular senescence is frequently associated with increased protein synthesis [7]. The increase in protein synthesis includes the excessive production of proinflammatory cytokines, called the senescence-associated secretory phenotype (SASP), which contributes to the induction of senescence [7,8]. The in vitro and in vivo markers of senescent cells include increased cell size; increased expression of p16INK4A, p21, p53, γH2AX, and SASP factors such as interleukin (IL)-6 and transforming growth factor-β1 (TGF-β1); and increased activity of senescence-associated β-galactosidase (SA-β-gal) [9,10]. 

Senolytic drugs can selectively remove senescent cells by blocking the senescence-associated antiapoptotic pathways via which senescent cells survive, despite their own SASP [11,12]. The first clinical trials of senolytics illustrated that the administration of dasatinib and quercetin (DQ) improved the physical function of idiopathic pulmonary fibrosis (IPF) patients [13]. Recently, several clinical trials with senolytics have efficiently and effectively proceeded in parallel [14]. Some studies reported that senolytics reverse the accumulation of senescent cells in the ovaries of obese individuals and cisplatin-induced ovarian injury [15,16]. Although one study reported that Cy induces the senescence of fibroblasts [17], it is unclear whether Cy induces cellular senescence in the ovaries and whether senolytics alleviate Cy-induced POI. 

Here, we hypothesized that Cy induces the senescence of granulosa cells in antral follicles and that accumulated senescent granulosa cells in ovaries worsen the microenvironment by secreting SASP factors, including TGF-β1, induce burn-out, and, finally, cause POI. To test our hypothesis, we first examined the expression of cellular senescence markers in primary cultured human granulosa-lutein cells (GLCs) treated with the Cy metabolite 4-hydroperoxy cyclophosphamide (4-HC) and granulosa cells of POI mice. Then, we tested the effects of DQ treatment on the expression of these markers and the SASP. We also examined the in vivo effect of DQ on the development of POI in a Cy-treated mouse model. Our results show that DQ is a promising agent to protect against Cy-induced ovarian damage.

## 2. Results

### 2.1. Cy induces Senescence of Human GLCs 

To determine whether Cy induces the senescence of granulosa cells, we exposed cultured human GLCs to an active metabolite of Cy, 4-HC, in vitro. Western blotting showed that the cellular senescence markers p16 (*p* < 0.01), p21 (*p* < 0.01), p53 (*p* < 0.01), and γH2AX (*p* < 0.05) were significantly upregulated in 4-HC-treated human GLCs (Figure 1). Immunostaining of SA-β-gal showed that 4-HC treatment increased the level of senescent cells (Appendix A). We also examined the effect of senolytics on the senescence of 4-HC-treated human GLCs. Human GLCs were pretreated with DQ for 24 h and then incubated with 4-HC. DQ abrogated the effects of 4-HC on cellular senescence markers (Figure 1). Thus, our in vitro assay indicates that senolytic drugs can prevent senescence of granulosa cells in POI patients. 

### 2.2. DQ Alleviate Senescence of Granulosa Cells in POI Model Mice

To better characterize cellular senescence in POI, we used a widely accepted Cy-induced POI mouse model (Figure 2A) [18]. We immunohistochemically analyzed the ovaries of mice in the control (*n* = 5), POI (*n* = 4), DQ (*n* = 5), and POI + DQ (*n* = 4) groups. Quantitative analysis of immunohistochemical staining showed that the cellular senescence markers p16 (*p* < 0.01), p21 (*p* < 0.01), p53 (*p* < 0.01), and γH2AX (*p* < 0.05) were significantly upregulated in the granulosa cells of antral follicles in the POI group compared with those in the control group (Figure 3). Cotreatment with DQ significantly decreased expression of p16 (*p* < 0.01), p21 (*p* < 0.05), p53 (*p* < 0.01), and γH2AX (*p* < 0.01). To assess the secretion of SASP factors, we evaluated TGF-β1. The serum concentration of TGF-β1 and its protein expression in granulosa cells of antral follicles were significantly increased in the POI group (*p* < 0.05 and *p* < 0.01, respectively), and these increases were significantly reversed in the POI+DQ group (*p* < 0.05 and *p* < 0.01, respectively; Figure 4). These findings indicate that the senescence of granulosa cells is induced in antral follicles of Cy-induced POI mice and this effect is abrogated by cotreatment with DQ.

### 2.3. DQ Partially Recover Burn-Out in POI Model Mice

To elucidate the effect of DQ on follicular development in POI, we investigated the number of follicles at all stages. Ovarian sections were assessed at 1 week after Cy treatment (Figure 2A). There was a significant loss of primordial (*p* < 0.01) and primary (*p* < 0.01) follicles and a significant increase in atretic follicles (*p* < 0.05 for number and *p* < 0.01 for percentage) in the POI group (Figure 5A–E,H). Ovarian follicles were categorized as dormant (primordial) and growing (primary, secondary, and antral) [19]. The reduction in dormant follicles was greater than that of early growing follicles in the POI group, which demonstrates that the induction of dormant follicles into the growing stage was activated, a phenomenon called burn-out (Figure 5G). However, this phenotype was partially alleviated in the POI+DQ group. The numbers of primordial (*p* < 0.01) and primary (*p* < 0.01) follicles were significantly higher, and the level of atretic follicles (*p* < 0.05 for number and *p* < 0.01 for percentage) was significantly lower in the POI+DQ group than in the POI group (Figure 5A–E,H). Consequently, the ratio of growing to dormant follicles tended to be lower in the POI+DQ group than in the POI group (*p* = 0.0760, Figure 5G). Ovarian sections also showed that there was no corpus luteum in ovaries in the POI group, and this effect was reversed in the POI+DQ group (*p* < 0.05, Figure 5F), which indicates that spontaneous ovulation was restored. To evaluate the ovulation capacity in response to ovulation induction, ovulated oocytes were counted after pregnant mare serum gonadotropin (PMSG)-human chorionic gonadotropin (hCG) injection on day 7 (control, *n* = 12; POI, *n* = 9; DQ, *n* = 11; and POI+DQ, *n* = 11, Figure 2B). The number of ovulated oocytes tended to be higher in the POI group than in the control group (*p* = 0.1352) and was significantly lower in the POI+DQ group than in the POI group (*p* < 0.05) (Figure 6A). According to the results shown in Figure 5, it was reasonable to assume that mice in the POI group ovulated more oocytes due to burn-out upon the short-term single ovulation induction protocol. Thus, we designed a long-term ovulation induction protocol for further verification. For the repeated PMSG/hCG injection protocol, mice were administered PMSG/hCG (control, *n* = 9; POI, *n* = 9; DQ, *n* = 9; and POI+DQ, *n* = 9) three times with a 2-week interval (Figure 2C). After the third ovulation induction, the number of oocytes was evaluated. The number of ovulated oocytes tended to be lower in the POI group than in the control group, which indicates that the reserve of follicles was exhausted in the POI group (Figure 6B). The number of ovulated oocytes was higher in the POI+DQ group than in the POI group (*p* = 0.0804). Our results indicate that the administration of Cy induces loss of the primordial follicle reserve and the administration of DQ partially rescues this effect.

### 2.4. DQ Improve Fertility of POI Model Mice

Given that burn-out was rescued in the POI+DQ group, we tested whether DQ improve the fertility of POI model mice. Mating with a wild-type male mouse was started on day 7 and repeated at 2 weeks after delivery (Figure 2D). Thus, 36 days (1 day for mating, 21 days for pregnancy, and 14 days for maternity leave) were designated as a mating cycle. The litter size in all groups declined after the seventh mating cycle (around 46 weeks old) due to reproductive aging (Figure 7). There is no difference in litter size up to the fifth mating cycles (control: 8.3 ± 1.3, POI 8.2 ± 1.2, POI+DQ 7.9 ± 1.2, mean ± SD/cycle). At the sixth mating cycle (around 41 weeks old), the litter size in the control and POI+DQ groups was comparable with that in previous mating cycles but was significantly smaller in the POI group than in the other groups (control: 8.7 ± 1.5, POI 5.8 ± 1.3, POI+DQ 8.0 ± 0.8, mean ± SD/cycle). These results indicate that single administration of Cy at a young age shortens reproductive life, and cotreatment with DQ rescues decreased fertility at older ages in a POI mouse model.

## 3. Discussion

The present study reports that expression of the cellular senescence markers p16, p21, p53, and γH2AX was increased in granulosa cells of antral follicles in Cy-induced POI model mice and 4-HC-treated human granulosa cells. Treatment with DQ abrogated Cy-induced expression of these markers. In addition, the serum concentration of TGF-β1, a SASP factor, and its protein expression in granulosa cells were significantly increased in POI model mice, and these increases were reversed by cotreatment with DQ. Administration of Cy induced burn-out, and this effect was partially rescued by cotreatment with DQ. Moreover, treatment with DQ significantly improved the ovulation rate and fertility of POI model mice in the long term.

We first tested whether Cy induces senescence of granulosa cells. Multiple signals can induce a cell to enter senescence, such as inflammation, radiation, and oxidative stress [8]. In senescent cells, the expression of markers of cell cycle arrest, apoptosis resistance, and the SASP are upregulated. There are several markers of cellular senescence, such as p16, p21, p53, γH2AX, IL-6, IL-8, IL-1α, SA-β-gal, and TGF-β1 [20]. We showed that, 4-HC, an active metabolite of Cy, upregulated the protein expression of p16, p21, p53, and γH2AX and the activity of SA-β-gal in human GLCs. This is consistent with the previous finding that Cy increases the expression of p16 and p53 and activity of SA-β-gal in cultured mouse granulosa cells [21]. Further confirming our hypothesis, we investigated cellular senescence in a Cy-induced POI mouse model. The expression of p16, p21, p53, and γH2AX was upregulated in the granulosa cells of antral follicles in these mice. The long-term persistence of senescent cells and their SASP disrupt tissue structure and function and have deleterious paracrine and systemic effects that cause fibrosis, inflammation, and a possible carcinogenic response [22]. Our study showed that the serum level of TGF-β1 and its protein expression in granulosa cells were increased in POI model mice, indicating that senescent granulosa cells secreted this SASP factor, although the elevated serum level of TGF-β1 did not rule out the possibility that Cy directly affects other cell types. Cellular senescence is induced upon ovarian injury caused by chemotherapies, including cisplatin and doxorubicin [15,23]. There are several molecular pathways behind Cy-induced cellular senescence. One study reported that it is induced by the lncRNA-Meg3-p53-p66Shc pathway in granulosa cells [21], and another study demonstrated that Cy induced cellular senescence via the activation of MAPK in fibroblasts [17]. The present study, together with these previous studies, strongly suggests that cellular senescence plays a crucial role in ovarian injury caused by chemotherapy.

Senolytics are therapeutics that remove senescent cells [24] and can disable the senescence-associated antiapoptotic pathway, which defends senescent cells against their apoptotic environment without affecting proliferating or quiescent differentiated cells [11,22]. Senolytics are new and promising therapeutic agents, not only for aging-related diseases but also for various inflammation-related diseases, such as osteoarthritis and COVID-19 [25,26]. DQ have been used in phase II clinical trials for chronic kidney disease [27] and IPF [13]. We treated human granulosa cells with DQ for 24 h before administration of 4-HC and administered DQ to mice for 3 consecutive days starting from the day of Cy administration. Upon administration of DQ, expression of the cellular senescence markers p16, p21, p53 and γH2AX was decreased in granulosa cells, both in vivo and in vitro. Furthermore, the serum level of TGF-β1 and its protein expression in granulosa cells were significantly lower in the POI+DQ group than in the POI group. These results indicate that senescent granulosa cells induced by Cy treatment are removed upon the administration of DQ and, consequently, the senescence load in the ovaries decreases. 

Two distinct mechanisms underlie the development of chemotherapy-induced POI. One is direct damage of DNA in oocytes of primordial follicles, while the other is the induction of a wave of activation of dormant follicles and resultant loss of the primordial follicle reserve, known as burn-out [19]. We observed burn-out in a POI mouse model. The ratio of growing to dormant follicles was significantly increased in these mice. In addition, the level of atretic follicles was significantly increased in the POI group. With the short-term single ovulation induction protocol, the number of ovulated oocytes was higher in the POI group than in the control group because Cy stimulated the activation of dormant follicles. On the other hand, with the long-term ovulation induction protocol, the number of ovulated oocytes was lower in the POI group than in the control group because Cy stimulated the loss of the primordial follicle reserve. We also tested the fertility of mice. Litter size was smaller in the POI group than in the other groups at the sixth mating cycle, showing that fertility was perturbed earlier in the former group and confirming that follicles were lost after the administration of Cy at a young age. However, the administration of DQ improved both ovulation induction and fertility. While DQ treatment rescues the defects of cisplatin-induced POI models, it does not improve the ovarian reserve or fertility in doxorubicin-induced POI models [15,23]. The mechanisms underlying gonadotoxicity differ according to the type of chemotherapy. Cisplatin and Cy activate burn-out, while doxorubicin causes DNA damage in primordial follicles [28]. These mechanistic differences may affect the efficacy of senolytics.

This study has some limitations. First, we focused on granulosa cells of follicles at later stages of development. One reason for this is that human primary granulosa cells available for in vitro experiments only serve as a model of granulosa cells in later-stage follicles. In addition, we expected that it would be difficult to determine whether the high expression of cellular senescence markers in primordial follicles is due to cellular senescence because these follicles highly express p16 and p21 to maintain quiescence [29]. Future studies utilizing follicles in vitro will clarify cellular senescence in dormant and earlier-stage follicles. The second limitation of our study is that we did not examine the mechanism by which Cy-induced senescent granulosa cells induce ovarian damage. We hypothesize that accumulated senescent granulosa cells in later-stage follicles worsen the microenvironment by secreting SASP factors, including TGF-β1, disturb the well-organized process of follicular recruitment and development, and stimulate burn-out. In a future study, the precise mechanism underlying Cy-induced POI should be elucidated to establish a strategy for preventing ovarian damage induced by Cy, the most gonadotoxic chemotherapy. Third limitation is that there is a difference in doses between our models and the real world for humans. Our doses of Cy and DQ were determined based on previous reports but this is higher than those in clinical trials for humans. There may be a dose-dependent bias in the results.

## 4. Materials and Methods

### 4.1. Human Specimens

GLCs and follicular fluid were aspirated from patients undergoing oocyte retrieval for in vitro fertilization at the University of Tokyo Hospital, Matsumoto Ladies Clinic, and Phoenix ART Clinic, in Tokyo, Japan.

### 4.2. Isolation and Culture of Human GLCs

GLCs were isolated as reported previously [30,31]. Follicular fluid was centrifuged at 1500 rpm at 20 °C for 10 min, and the pellet was resuspended in phosphate-buffered saline (PBS) containing 0.2% hyaluronidase (Sigma-Aldrich, St. Louis, MO, USA) and incubated in a 37 °C water bath for 30 min. The suspension was layered over Ficoll-Paque (GE Healthcare, Chicago, IL, USA) and centrifuged at 400× *g* for 30 min. GLCs were collected from the interface, washed with PBS, and cultured in Dulbecco’s Modified Eagle Medium F-12 (Thermo Fisher Scientific, Waltham, MA, USA) containing 10% fetal bovine serum (Sigma-Aldrich) and antibiotics (100 U/mL penicillin, 0.1 mg/mL streptomycin, and 250 ng/mL amphotericin B; Sigma-Aldrich). GLCs were adjusted to a density of 5 × 105 cells/mL and cultured in 12-well plates. All GLCs were precultured for 3–5 days prior to treatment at 37 °C in a humidified atmosphere containing 5% CO_2_.

### 4.3. Treatment of Human GLCs

To evaluate the effect of Cy on activation of cellular senescence, GLCs were treated with an active metabolite of Cy, 4-HC [32,33] (Toronto Research Chemicals, Toronto, ON, Canada), at concentrations of 0.001–100 μM for 24 h. These conditions were chosen based on previous studies that used human cells or animal GLCs [34,35,36]. Treatment with 10 μM 4-HC for 24 h exerted the maximum effect without changing cellular morphology. Hence, stimulation with 10 μM 4-HC for 24 h was performed in subsequent experiments. To evaluate the effect of DQ on Cy-induced cellular senescence and expression of senescence markers, GLCs were preincubated with 1 nM dasatinib (Sigma-Aldrich) and 10 μM quercetin (Sigma-Aldrich) for 24 h and then treated with 4-HC. To determine the optimal concentrations of DQ, dose determination experiments were conducted with a combination of 1–1000 nM dasatinib and 1–20 μM quercetin according to previous studies [15,37]. For combinational treatment, 1 nM dasatinib and 10 μM quercetin were the lowest concentrations that exerted a significant effect (Appendix A).

### 4.4. POI Animal Model

Eight-week-old male and 7–12-week-old female C57BL/6J mice were provided by Japan SLC Inc. (Hamamatsu, Japan). Mice were maintained under specific pathogen-free conditions and a 12 h light/12 h dark cycle with ad libitum provision of water and feed. After adapting to their surroundings for 1 week, mice were divided into four groups (control, DQ, POI, and POI+DQ). The experimental design is shown in Figure 2A. The control and DQ groups were intraperitoneally (i.p.) administered 0.1 mL PBS and the POI and POI+DQ groups were i.p. administered 0.1 mL Cy (75 mg/kg; Tokyo Chemical Industry Co., Ltd., Tokyo, Japan) on day 0. The DQ and POI+DQ groups were orally administered 0.1 mL of a mixture of dasatinib (5 mg/kg, Sigma-Aldrich) and quercetin (50 mg/kg, Sigma-Aldrich) dissolved in 10% polyethylene glycol 400 (PEG400; Fujifilm, Tokyo, Japan), and the control and POI groups were administered an equal volume of 10% PEG400 from day 0 to day 2. The first administration of DQ was in the morning, about 8~10 hours before administration of Cy. The procedures and doses of Cy, dasatinib, and quercetin were determined as previously described [18,38,39]. On day 7, mice were euthanized under isoflurane anesthesia (Viatris, Canonsburg, PA, USA), and blood, ovary, and oviduct samples were collected. 

#### 4.4.1. Ovulation Induction

A flowchart of the ovulation protocol is shown in Figure 2B,C. In the short-term ovulation induction protocol (Figure 2B), at 7 days after administration of Cy, mice were i.p. injected with 10 IU PMSG (Asaka Pharmaceutical, Tokyo, Japan) followed by 10 IU hCG (Mochida Pharmaceutical, Tokyo, Japan) 48 h later. At 12–14 h after administration of hCG, oocytes were isolated from oviducts under an optical microscope (Olympus IX70; Olympus, Tokyo, Japan) [40]. The number of oocytes was recorded. In the long-term ovulation induction protocol (Figure 2C), PMSG-hCG administration was repeated three times with a 2-week interval. Oocytes were collected after the third cycle of superovulation induction.

#### 4.4.2. Mating Test

A flowchart of the mating protocol is shown in Figure 2D. At 1 week after administration of Cy, one or two female mice were mated with an 11-week-old male mouse. Pregnant mice were allocated to an independent cage until delivery. The number of pups was recorded. At 2 weeks after delivery, mating was repeated until the female mouse failed to conceive upon continuous mating for more than 1 month, the general condition of the female mouse prohibited pregnancy, or postmenopausal and aging symptoms appeared, such as closure of the vagina and skin ulcer-like changes. The duration of the mating test was 308 days in total.

### 4.5. Western Blotting

GLCs were lysed in PhosphoSafe Extraction Reagent (Merck, Darmstadt, Germany) and centrifuged at 16,000× *g* for 5 min at 4 °C to remove insoluble material. Supernatants were recovered, and protein concentrations were measured using the Bio-Rad Protein Assay (Bio-Rad Laboratories, Hercules, CA, USA). Equivalent amounts of denatured protein were subjected to SDS-PAGE and then electrophoretically transferred to polyvinylidene difluoride membranes using the Trans-Blot Turbo Transfer System (Bio-Rad Laboratories). After blocking with 5% skim milk dissolved in Tris-buffered saline containing 0.1% Tween-20 at room temperature for 1 h, the membranes were probed with anti-p16 (1:500, RRID: AB_628067; Santa Cruz Biotechnology Inc., Dallas, TX, USA), anti-p21 (1:1000, RRID: AB_2077682; Proteintech, Tokyo, Japan), anti-p53 (1:5000, RRID: AB_2881401, Proteintech), anti-phospho-histone H2A.X (1:400, RRID: AB_2118009; Cell Signaling Technology, Danvers, MA, USA), and anti-β-actin (1:10,000, RRID: AB_476697, Proteintech) antibodies overnight at 4 °C and then incubated with secondary antibodies (1:2000; anti-rabbit, RRID: AB_2099233 or anti-mouse, RRID: AB_330924; Cell Signaling Technology) at room temperature for 1 h. Images were acquired using ECL Plus Western blotting detection reagents (GE Healthcare) on an ImageQuant LAS 4000 mini luminescent image analyzer (GE Healthcare). β-actin was used as a loading control. The immunoblot procedure was repeated at least three times. Band intensities were quantified using ImageJ software version 1.53a (RRID: SCR_003073; National Institutes of Health, Bethesda, MD, USA) [41]. 

### 4.6. Histology and Immunohistochemistry

Mouse ovaries were fixed in 10% neutral-buffered formalin, embedded in paraffin, and serially sectioned every 5 μm from the maximum slide to the bilateral side. Every fifth section was stained with hematoxylin and eosin. Twenty-six stained slides were assessed to represent the whole ovary. Follicles were classified and counted to evaluate the effect of Cy on ovarian follicles and the protective effects of DQ [42]. Atretic follicles were counted when attenuation of the granulosa cell layer, shrinkage, or degeneration of oocyte nuclei was observed [15]. Only follicles with clearly visible oocyte nuclei were counted. Ovarian sections were immunostained with anti-p16 (1:500), anti-p21 (1:1000), anti-p53 (1:5000), anti-phospho-histone H2A.X (1:400), and anti-TGF-β1 (1:1000, RRID: AB_10562492; Abcam, Cambridge, UK) antibodies using an EnVision+ Dual Link System/HRP (DAB) Kit (Dako, Tokyo, Japan). Isotype-specific IgG served as a negative control. Antigen retrieval was performed using target retrieval solution (Dako). Immunohistochemistry was independently performed at least three times using identical samples.

### 4.7. SA-β-gal Staining

SA-β-gal staining of GLCs was performed as reported previously [43]. GLCs were prewashed with PBS, fixed with 8 mL of 0.5% glutaraldehyde solution (FUJIFILM Wako Pure Chemical Corporation, Osaka, Japan) and 0.2 mL of 2% formaldehyde solution (FUJIFILM Wako Pure Chemical Corporation), dissolved in 1.8 mL PBS (pH = 7.4), and then stained with staining solution. The staining solution comprised X-gal (5-bromo-4-chloro-3-indolyl-β-D-galactopyranoside, FUJIFILM Wako Pure Chemical Corporation), MgCl2, and potassium ferrocyanide (FUJIFILM Wako Pure Chemical Corporation) dissolved in PBS (pH = 6.0). SA-β-gal staining was examined using an Olympus BX50 fluorescence microscope (Olympus).

### 4.8. ELISA

To determine the concentration of TGF-β1, a SASP factor, in mouse serum, blood samples were collected from each group of mice (*n* = 10) and centrifuged. The separated serum was stored at −80 °C until assayed. The serum concentration of TGF-β1 was measured using an ELISA kit (R&D Systems, Minneapolis, MN, USA).

### 4.9. Statistical Analysis

All statistical analyses were performed with Prism software version 9.5.0 (GraphPad Software, LLC, San Diego, CA, USA). All data are shown as mean ± standard deviation (SD). Data were analyzed using Student’s *t*-test for paired comparisons. *p* < 0.05 was considered statistically significant. All experiments were independently repeated at least three times.

## 5. Conclusions

We demonstrated that senescence is activated in granulosa cells upon Cy-induced POI. The accumulation of senescent cells in ovaries contributes to infertility. The administration of senolytics, which remove senescent cells, partially recovers ovarian function. Targeting senescent cells with senolytics, such as DQ, might be a promising strategy to protect against Cy-induced ovarian damage.

## Figures and Tables

**Figure 1 ijms-24-17193-f001:**
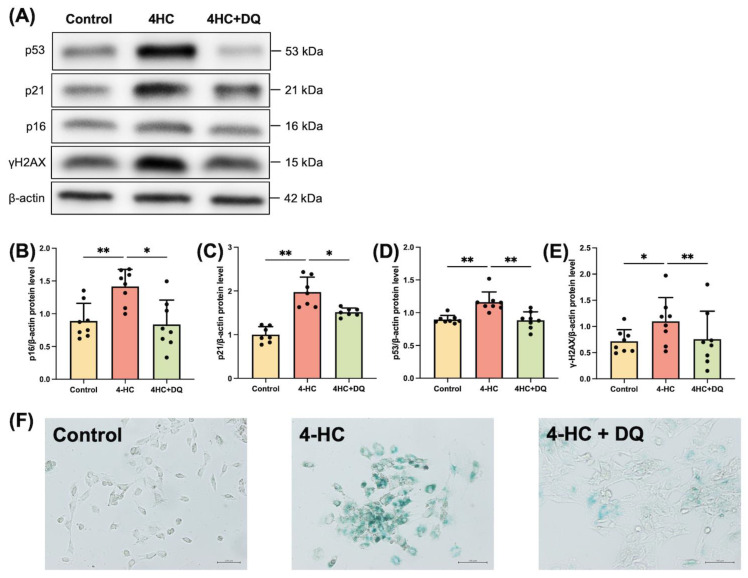
Cy induces senescence of cultured human GLCs. Human GLCs were preincubated with DQ or vehicle for 24 h and then treated with 4-HC, an active metabolite of Cy, or vehicle. (**A**) The protein expression levels of p16, p21, p53, and γH2AX in human GLCs was analyzed by western blotting. β-actin was used as a loading control. Representative blots are shown. (**B**–**E**) Quantitative analysis of western blotting. Values represent mean ± SD. Student’s *t*-test was used for statistical analysis. * *p* < 0.05 and ** *p* < 0.01 compared with the 4-HC-treated group. (**F**) Human GLCs were preincubated with 1 nM dasatinib and 10 μM quercetin for 24 h and then treated with 10 μM 4-HC. Light microscopic images of SA-β-gal staining are shown. Scale bars indicate 200 μm.

**Figure 2 ijms-24-17193-f002:**
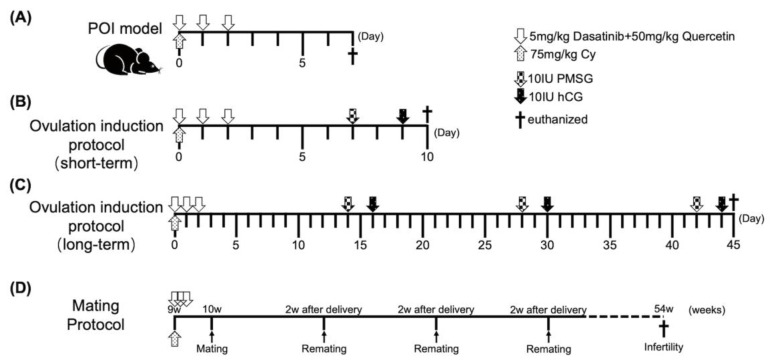
Experimental design for POI model. (**A**) POI model with DQ treatment. Eight-week-old mice were i.p. injected with 0.1 mL PBS (control and DQ groups) or 0.1 mL PBS containing Cy (75 mg/kg, POI and POI+DQ groups) on day 0. Mice were orally administered 0.1 mL of 10% PEG400 (control and POI groups) or 0.1 mL of 10% PEG400 containing dasatinib (5 mg/kg) and quercetin (50 mg/kg) (DQ and POI+DQ groups) from day 0 to day 2. On day 7, mice were euthanized, and blood, ovary, and oviduct samples were collected. (**B**) Short-term ovulation induction protocol. At 7 days after administration of Cy, mice were i.p. injected with 10 IU PMSG, followed by 10 IU hCG 48 h later. At 12–14 h after administration of hCG, oocytes were collected. (**C**) Long-term ovulation induction protocol. PMSG-hCG administration was repeated three times with a 2-week interval. Oocytes were collected after the third ovulation induction. (**D**) Mating protocol. Female mice were mated with a male mouse at 1 week after administration of Cy. At 2 weeks after delivery, female mice were remated with a male mouse up to 54 weeks of age. The litter size of each delivery was recorded to evaluate fertility of each group.

**Figure 3 ijms-24-17193-f003:**
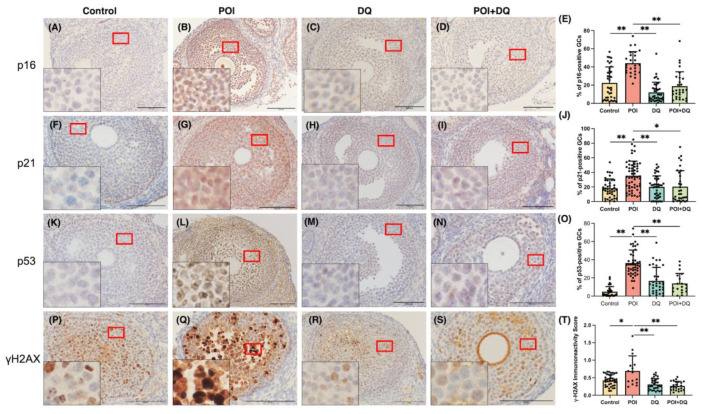
DQ alleviate cellular senescence in Cy-induced POI model mice. Immunohistochemical analysis was performed of antral follicles of ovaries from mice in the control (*n* = 5), POI (*n* = 4), DQ (*n* = 5), and POI+DQ (*n* = 4) groups. (**A**–**D**,**F**–**I**,**K**–**N**,**P**–**S**) Cross-sections of ovaries were stained with an anti-p16 (**A**–**D**), anti-p21 (**F**–**I**), anti-p53 (**K**–**N**), or anti-H2AX (**P**–**S**) antibody and counterstained with hematoxylin. (**E**,**J**,**O**,**T**) Quantitative analysis of immunohistochemical staining. Three to five follicles from each mouse were evaluated. The black box at the bottom left shows the enlarged image of GC staining. Values represent mean ± SD. Scale bars indicate 100 μm. Student’s *t*-test was used for statistical analysis. * *p* < 0.05 and ** *p* < 0.01 compared with the POI group.

**Figure 4 ijms-24-17193-f004:**
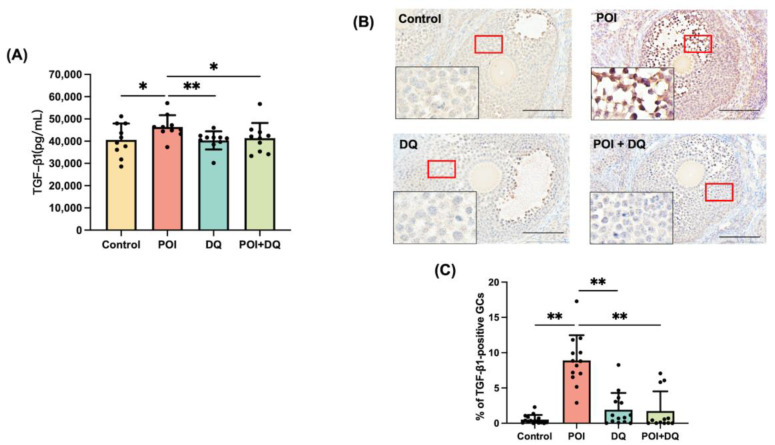
DQ alleviate secretion of TGF-β1 in POI model mice. (**A**) Plasma concentrations of TGF-β1 were measured in 10-week-old mice at 1 week after Cy treatment with or without DQ (*n* = 10 per group). (**B**) Cross-sections of ovaries were stained with an anti-TGF-β1 antibody and counterstained with hematoxylin. (**C**) Quantitative analysis of immunohistochemical staining. Three to five follicles from each mouse were evaluated. The black box at the bottom left shows the enlarged image of GC staining. Values represent mean ± SD. Scale bars indicate 100 μm. Student’s *t*-test was used for statistical analysis. * *p* < 0.05 and ** *p* < 0.01 compared with the POI group.

**Figure 5 ijms-24-17193-f005:**
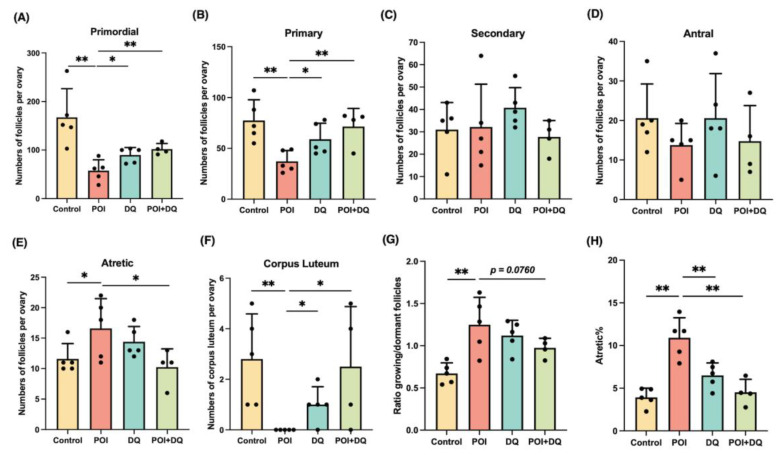
DQ partially rescue burn-out in POI model mice. Follicles in ovaries were evaluated at 1 week after treatment with Cy (POI, *n* = 5 and POI+DQ, *n* = 4) or PBS (control, *n* = 5 and DQ, *n* = 5) with or without DQ. (**A**–**E**) Counts of primordial, primary, secondary, antral, and atretic follicles in each group. (**F**) Number of corpora lutea in each group. (**G**) Ratio of growing (primary, secondary, and antral) to dormant (primordial) follicles in each group. (**H**) Percentage of atretic follicles in each group. Data represent mean ± SD. Student’s *t*-test was used for statistical analysis. * *p* < 0.05 and ** *p* < 0.01 compared with the POI group.

**Figure 6 ijms-24-17193-f006:**
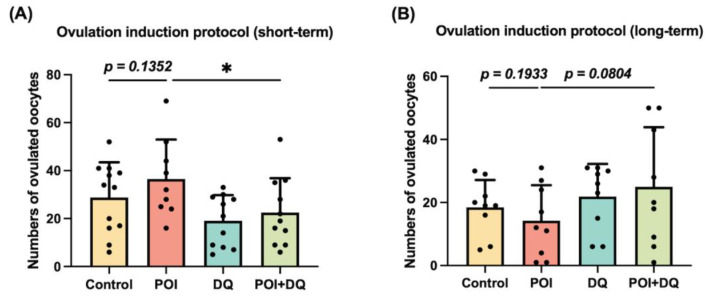
Effects of DQ on short- and long-term ovulation induction in POI model mice. (**A**) Model mice were i.p. injected with PMSG at 7 days after Cy administration, followed by hCG 48 h later (control, *n* = 12; POI, *n* = 9; DQ, *n* = 11; and POI+DQ, *n* = 11). Ovulated oocytes were collected at 12–14 h after hCG injection. The numbers of oocyte in each group were evaluated. (**B**) For the long-term ovulation induction protocol, PMSG/hCG were injected three times with a 2-week interval. After the third ovulation induction, the numbers of oocytes were evaluated. Values represent mean ± SD. Student’s *t*-test was used for statistical analysis. * *p* < 0.05 compared with the POI group.

**Figure 7 ijms-24-17193-f007:**
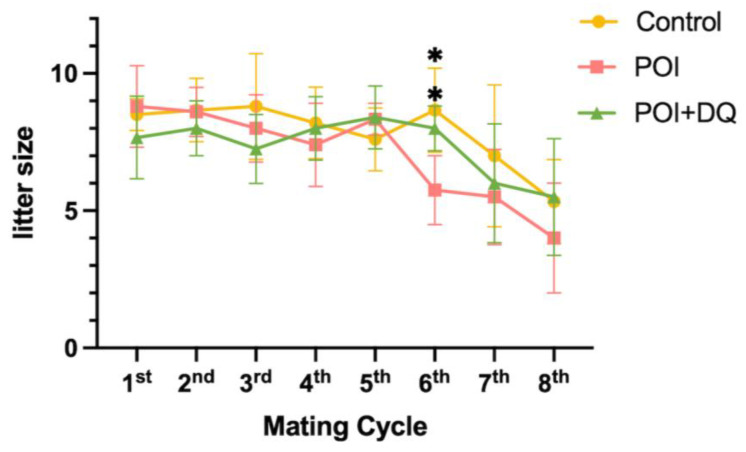
DQ improve fertility of POI model mice. Female mice were mated with a male mouse at 7 days after Cy administration (control, *n* = 5; POI, *n* = 6; and POI+DQ, *n* = 6). Thirty-six days (1 day for mating, 21 days for pregnancy, and 14 days for maternity leave) were designated a mating cycle. Litter size was recorded every cycle and compared between the groups. Values represent mean ± SD. Student’s *t*-test was used for statistical analysis. * *p* < 0.05 compared with the POI group.

## Data Availability

All datasets are included in the manuscript and Appendix A. Raw data files are available upon reasonable request.

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
