# Peer review of "The Role of Cellular Senescence in Cyclophosphamide-Induced Primary Ovarian Insufficiency"

_ijms, 2023, doi:10.3390/ijms242417193_

Round 1
Reviewer 1 Report
Comments and Suggestions for Authors
The manuscript entitled “The role of cellular senescence in cyclophosphamide-induced primary ovarian insufficiency” provides new insights that, cellular senescence plays critical roles in Cy-induced POI, and targeting senescent cells with senolytics, such as DQ, might be a promising strategy to protect against Cy-induced ovarian damage. It needs major revision to make the suitable for publication.
1. In figure 1, authors should confirm Cell cycle analysis after treatment with Cy in cultured human GLC cells.
2. Determining β-galactosidase activity would be a great marker of cellular senescence so I recommend including this in manuscript.
3. In this this study authors have a discussion about the molecular pathway behind Cy induced senescence.
4. Also include a graphical abstract for better understanding.
Reviewer 2 Report
Comments and Suggestions for Authors
In this manuscript, Zixin Xu et al showed that the role of cellular senescence in cyclophosphamide induced primary ovarian insufficiency. These findings are potentially interesting. The manuscript could be further strengthened with a few additional things denoted below.
1. The authors need to explain more about ovarian cancer in introduction part.
2. Please explain more detail about figure 7 data in result part.
3. What exactly are the changes in protein expression of p16, p21, p53, and H2AX related?
Comments on the Quality of English LanguageNone.
Reviewer 3 Report
Comments and Suggestions for Authors
In this manuscript, the authors investigated cellular senescence-mediated primary ovarian insufficiency (POI) when treated with a chemotherapy drug cyclophosphamide (Cy), and the effect of senolytics therapy in rescuing Cy-induced ovarian damage, using both human cells and mouse model. They found upregulated cellular senescence markers with Cy treatment which could be ameliorated by senolytics in both in vitro and in vivo studies. They further demonstrated that senolytics treatment rescued decreasing numbers of primordial and primary follicles caused by Cy, and improved the response to ovulation induction and fertility in mice. Altogether, their results suggested the important roles of cellular senescence in Cy-induced POI and senolytics could be a promising strategy to deal with this chemotherapy side effect. Overall, the manuscript was organized and well written, and the new insights revealed by it can be of potential interest to a broad audience. Some minor comments to consider are:
1. In the mouse model, 75 mg/kg Cy was administered with or without 5 mg/kg dasatinib and 50 mg/kg quercetin, which appear to be relatively high doses. How does this dose level compare with actual chemotherapy and senolytics clinical trials? Will lower doses of Cy still cause POI via cellular senescence? Will lower dose DQ treatment still be helpful to abrogate cellular senescence? It is beneficial to include more discussions to help readers better understand differences between real world cases and the design of current study, and any potential dose-dependent bias of the results.
2. In human GLC experiment, cells were pretreated with DQ while in the mouse study it was co-dosed with Cy. Is senolytics functioning more in a protective way or does it also help even after the ovarian damage?
3. In Figure 3, four or five replicates were included in each group. However, in panels E, J, O, T a lot more data points were shown. Is this because multiple IHC images or areas from one mouse were used for quantitation? Please clarify in the figure caption. Same comment for Figure 4C.
Reviewer 4 Report
Comments and Suggestions for Authors
This study evaluated the potential role of cell senescence in Cy-induced ovarian insufficiency through the clearance of Sns cells. Overall, this study was well-designed and the experiments performed rigorously with proper controls. Please see below my minor concerns.
1, The blots for p16 and p21 showing in Fig 1A have low resolution. Please provide high-resolution images.
2, The schematics in Fig2 should be combined with Fig3.
3, Zoom-in images are required for Fig 3A-N. It's hard to tell positive cells in the existing images.
4, Most of the sections in DQ and POI+DQ groups are cracked. Is this due to just bad section or the effects of DQ administration?
Comments on the Quality of English Language
This study evaluated the potential role of cell senescence in Cy-induced ovarian insufficiency through the clearance of Sns cells. Overall, this study was well-designed and the experiments performed rigorously with proper controls. Please see below my minor concerns.
1, The blots for p16 and p21 showing in Fig 1A have low resolution. Please provide high-resolution images.
2, The schematics in Fig2 should be combined with Fig3.
3, Zoom-in images are required for Fig 3A-N. It's hard to tell positive cells in the existing images.
4, Most of the sections in DQ and POI+DQ groups are cracked. Is this due to just bad section or the effects of DQ administration?
Round 2
Reviewer 1 Report
Comments and Suggestions for Authors
Accept in current form
Author Response
Thank you for reviewing the revised manuscript.